CAF-derived miR-642a-3p supports migration, invasion, and EMT of hepatocellular carcinoma cells by targeting SERPINE1

Zhang Shuo 1
Cao Gang 2
Shen Shuijie 3
Wu Yu 3
Tan Xiying 4
Jiang Xiaoyan jxy20231973@163.com 1
1 Department of Pharmacy, Nantong Hospital of Traditional Chinese Medicine , Nantong , China
2 Office of the Dean, Nantong Maternal and Child Health Care Hospital , Nantong , China
3 Department of Science and Education, Nantong Hospital of Traditional Chinese Medicine , Nantong , China
4 Department of Pharmacy, Affiliated Hospital of Nanjing University of Chinese Medicine , Nanjing , China
Uversky Vladimir
Electronic publication date: 2024 Nov 11
Publication date: 2024
Volume: 12
Electronic Location ID: e18428
Received 2024 Jun 7; Accepted 2024 Oct 8
Copyright: ©2024 Zhang et al.
Copyright year: 2024
Copyright holder: Zhang et al.
License: This is an open access article distributed under the terms of the Creative Commons Attribution License, which permits unrestricted use, distribution, reproduction and adaptation in any medium and for any purpose provided that it is properly attributed. For attribution, the original author(s), title, publication source (PeerJ) and either DOI or URL of the article must be cited.
License URL: https://creativecommons.org/licenses/by/4.0/

Keywords: Hepatocellular carcinoma, Cancer-associated fibroblasts, miR-642a-3p, SERPINE1, Invasion, EMT

Funding: National Natural Science Foundation of China No. 82104408 No. 82003961 Scientific Research Project of Jiangsu Provincial Health Commission No. Z2021081 This work was supported by the National Natural Science Foundation of China (No. 82104408, No. 82003961) and the Scientific Research Project of Jiangsu Provincial Health Commission (No. Z2021081). The funders had no role in study design, data collection and analysis, decision to publish, or preparation of the manuscript.

==============================
Background

Cancer-associated fibroblasts (CAFs) and hepatocellular carcinoma (HCC) cells interact to promote HCC progression, but the underlying mechanisms remain unclear. Serpin family E member 1 (SERPINE1) has conflicting roles in HCC, and microRNAs (miRNAs) are known to regulate tumor progression through intercellular communication. Therefore, we investigated the potential involvement of miRNA/SERPINE1 axis in crosstalk between CAFs and HCC cells.

Methods

In this study, candidate miRNAs targeting SERPINE1 3′ UTR were predicted using multiple miRNA databases. The miRNAs and SERPINE1 mRNA expression in Huh7 cells was assessed after co-culture with CAFs using RT-qPCR. Huh7 cell proliferation and invasion were detected after SERPINE1 siRNA. The functions of the CAF-derived miR-642a-3p/SERPINE1 axis in HCC cells were examined using CCK-8, wound healing, transwell assays, western blot, and dual-luciferase reporter assays. Moreover, a orthotopic xenograft model was used to investigate the contribution of miR-642a-3p knockdown in HCC.

Results

SERPINE1 mRNA expression decreased, while miR-642a-3p expression increased in Huh7 cells co-cultured with CAFs. SERPINE1 knockdown enhanced Huh7 cell proliferation and invasion as well as miR-642a-3p expression. miR-642a-3p overexpression promoted migration, invasion, and epithelial-mesenchymal transition (EMT) in Huh7 cells by targeting SERPINE1, while miR-642a-3p knockdown yielded the opposite effect. Rescue experiments confirmed that SERPINE1 knockdown attenuated the inhibitory effects of miR-642a-3p knockdown on migration, invasion, and EMT in Huh7 cells. Importantly, miR-642a-3p knockdown suppressed growth and EMT in orthotopic liver tumors.

Conclusion

CAF-derived miR-642a-3p/SERPINE1 axis facilitated migration, invasion, and EMT in the HCC cells, suggesting miR-642a-3p/SERPINE1 axis can be a potential therapeutic target for HCC.

Introduction

Liver cancer is a prevalent malignant neoplasm ranking as the fourth leading cause of cancer-related mortality (Ghafouri-Fard et al., 2021). Globally, in 2020, the incidence of new liver cancer cases and associated deaths was 9.5 and 8.7 per 100,000 individuals, respectively, with both rates exhibiting annual increases (Rumgay et al., 2022). Hepatocellular carcinoma (HCC) is the most common primary liver malignancy and constitutes approximately 80–90% of primary liver cancer (Ladd et al., 2024; Phoolchund & Khakoo, 2024; Sankar et al., 2024). The primary cause of HCC is chronic liver disease, including cirrhosis and chronic hepatitis (Sankar et al., 2024). Alcohol usage, non-alcoholic steatohepatitis (NASH), hepatitis B virus (HBV), and hepatitis C virus (HCV) can all result in chronic hepatitis (Phoolchund & Khakoo, 2024; Sankar et al., 2024). Despite significant advancements in HCC treatment over recent decades, early detection remains challenging. Consequently, more than 70% of patients are diagnosed at advanced disease stages, resulting in a markedly low five-year survival rate following surgical intervention (Hou et al., 2022; Vogel et al., 2022; Zhou & Song, 2021). Furthermore, chemotherapy resistance and recurrent disease are major contributors to the poor prognosis of HCC patients (Ganesan & Kulik, 2023; Hou et al., 2022). Consequently, the identification of biomarkers associated with HCC initiation and progression is crucial for enhancing prevention, diagnostic accuracy, therapeutic efficacy, and prognostication (Lv & Sun, 2024).

During tumorigenesis, a heterogeneous population of cells congregates around cancer tissues, forming a distinctive microenvironment known as the tumor microenvironment (TME) (Basak et al., 2023; Hoekstra et al., 2024; Yu, Huang & Guo, 2024). These cells are recruited to fulfill pro-tumorigenic functions, enabling cancer cells to evade immune surveillance and establish a tumor niche (Hoekstra et al., 2024; Yasuda & Wang, 2024; Yu, Huang & Guo, 2024). Cancer-associated fibroblasts (CAFs), a predominant component of the TME, exert a multitude of oncogenic effects within tumor tissues, including alterations in tumor metabolism and immune reprogramming, facilitation of immune evasion, enhancement of drug resistance, and modulation of the TME (Arpinati, Carradori & Scherz-Shouval, 2024; Jing et al., 2024; Ye et al., 2024). Through the secretion of growth factors, immunomodulatory molecules, and extracellular matrix proteins, CAFs remodel the extracellular matrix and TME, thereby promoting metastasis, immune escape, and therapeutic resistance in tumors (Arpinati, Carradori & Scherz-Shouval, 2024; Biffi & Tuveson, 2021; Helms, Onate & Sherman, 2020; Miyai et al., 2020). Consequently, CAFs have emerged as a focal point for clinical and preclinical investigations (Caligiuri & Tuveson, 2023). In HCC, activated CAFs interact with HCC cells, expressing various pro-proliferative and pro-invasive factors, thereby creating a permissive microenvironment for HCC cell proliferation, growth, invasion, and migration (Schneider et al., 2024; Shang et al., 2024). Moreover, characterizing distinct CAF clusters has demonstrated prognostic value in HCC, offering a novel therapeutic approach (Yu et al., 2022).

In HCC, CAFs stimulate serpin family E member 1 (SERPINE1) expression in tumor-associated macrophages (TAMs), thereby promoting the malignant progression of HCC cells through epithelial-mesenchymal transition (EMT) (Chen et al., 2021). As a member of the serine protease inhibitor family, SERPINE1 is a critical regulator of extracellular matrix remodelling (Higgins et al., 2019; Kong et al., 2021). This protein participates in diverse physiological processes, including metabolism, inflammation, angiogenesis, cancer, and aging (Sillen & Declerck, 2021; Wang et al., 2023a). SERPINE1 interacts with biological ligands, such as vitronectin and cell surface receptors, to engage in pericellular proteolysis, tissue remodeling, and cell migration (Sillen & Declerck, 2021). SERPINE1 has been implicated in multiple facets of cancer progression, including proliferation, migration, invasion, EMT, angiogenesis, and drug resistance (Nagy, Munkácsy & Győrffy, 2021; Su et al., 2024a; Teng et al., 2021). Overexpressed in various cancers, including gastric (Chen et al., 2022; Tutunchi et al., 2021), lung (Hong et al., 2022; Setiawan et al., 2024), and colon (Wang et al., 2023b) cancers, SERPINE1 is classified as a pan-oncogene and is associated with poor prognosis (Ju et al., 2024). However, the functional role of SERPINE1 in HCC remains controversial (Jin et al., 2020). While some studies report high SERPINE1 expression in HCC, promoting tumor progression (Zhang et al., 2021), others suggest a tumor-suppressive role for SERPINE1, with overexpression inhibiting HCC cell invasion (Wang et al., 2016). This complexity underscores the need for further investigation into the multifaceted functions of SERPINE1 in HCC.

Non-coding RNAs (ncRNAs) play a pivotal role in controlling cell communication within TME (Szymanowska et al., 2023). They regulate tumor cell proliferation, apoptosis, metastasis, and drug resistance, and are therefore considered as potential cancer markers (Chakrabortty et al., 2023; Iaccarino & Klapper, 2021). As a subset of ncRNAs, microRNAs (miRNAs) are single strand molecules with 20–24 nucleotides that bind to mRNA to control the expression of post-transcriptional genes (Chakrabortty et al., 2023). Dysregulation of miRNA expression is associated with cancer progression (Chakrabortty et al., 2023; Parsa-Kondelaji et al., 2023). While aberrant expression of many miRNAs, including miR-22 (Hu et al., 2023) and miR-17-5p (Zhou et al., 2024), has been linked to HCC progression, the involvement of miRNAs in CAFs-HCC communication remains relatively unexplored (Su et al., 2024b).

The purpose of this study was to investigate the role of CAF-derived miRNA on HCC cells through regulating SERPINE1. First, we screen candidate miRNAs targeting SERPINE1 through multiple miRNA databases (StarBase, miRwalk, TargetScan, and miRDB). Subsequently, we analyzed whether CAFS-derived miR-642a-3p targeted SERPINE1 via co-culture of CAFs and HCC cells, real-time quantitative PCR (RT-qPCR), and dual-luciferase reporter assays. Finally, the functional significance of miR-642a-3p/SERPINE1 axis in HCC cells was explored in vitro and in vivo to identify novel therapeutic targets for HCC.

Materials and Methods

Cell culture

Huh7 cells (a human hepatoma cell line) and human hepatocellular CAFs were obtained from Jiangsu KeyGEN Biotechnology Co., Ltd. (KeyGEN BioTECH, Nanjing, China) and Shanghai Fusheng Industrial Co., Ltd. (Shanghai, China), respectively. The cells were cultured in Dulbecco’s Modified Eagle’s Medium (DMEM) (KeyGEN BioTECH) supplemented with 10% fetal bovine serum (FBS) (Gibco, USA) and 1% penicillin-streptomycin (P/S) (KeyGEN BioTECH) in a humidified incubator at 37 °C with 5% CO2.

Huh7 cells and CAFs were co-cultured in a transwell system. CAFs were seeded in the upper chamber of the transwell, while Huh7 cells were seeded in the lower chamber. A transwell without CAFs served as the control. Co-culture was maintained for 48 h.

MiRNAs bioinformatics prediction

The miRNAs binding to SERPINE1 3′ untranslated region (UTR) were predicted using the starBase (https://rnasysu.com/encori/), miRwalk (http://mirwalk.umm.uni-heidelberg.de/), TargetScan (https://www.targetscan.org/vert_80/), and miRDB (https://mirdb.org/) databases. The predicted miRNAs were intersected by Venn diagram.

Cell transfection

Huh7 cells were transfected with miR-642a-3p mimics, miR-642a-3p inhibitor, or SERPINE1 siRNA (si-SERPINE1) (General Biology, Anhui, China) for 48 h using Lipofectamine™ 3000 transfection reagent (Invitrogen, USA) according to the manufacturer’s protocol. Briefly, 125 µL Opti-MEM medium containing 100 pmol SERPINE1 siRNA (20 µM) or miR-642a-3p mimics/inhibitor (20 µM) and 5 µL P3000™ reagent was gently mixed. Similarly, 125 µL Opti-MEM medium and 3.75 µL Lipofectamine™ 3000 reagent were combined. The diluted Lipofectamine™ 3000 reagent was then added to the diluted siRNA mixture, gently mixed, and incubated at room temperature for 10-15 min. Once the cell confluence reached 70–80% in the 6-well plate, 250 µL of the transfection mixture was added, and the cells were cultured at 37 °C for 48 h. Negative control groups included cells transfected with miR-642a-3p mimics negative control (mimics NC), miR-642a-3p negative inhibitor (inhibitor NC), or siRNA negative control (si-NC). Each experimental group was performed in triplicate. The sequences of the miR-381-3p mimics, inhibitor, SERPINE1 siRNA, and respective negative controls are provided in Table 1.

Table 1 The sequences of the miR-642a-3p mimics, inhibitor, SERPINE1 siRNA, and respective NCs.

Name	Sequence (5′ → 3′)	
miR-642a-3p mimics	F: AGACACAUUUGGAGAGGGAACC	
R: GGUUCCCUCUCCAAAUGUGUCU	
mimics NC	F: UCACAACCUCCUAGAAAGAGUAGA	
R: UCUACUCUUUCUAGGAGGUUGUGA	
miR-642a-3p inhibitor	GGUUCCCUCUCCAAAUGUGUCU	
Inhibitor NC	UCUACUCUUUCUAGGAGGUUGUGA	
SERPINE1 siRNA#1	F: GGAAAGGAGCCGUGGACCATT	
R: UGGUCCACGGCUCCUUUCCTT	
SERPINE1 siRNA#2	F: CGACAUGUUCAGACAGUUUTT	
R: AAACUGUCUGAACAUGUCGTT	
SERPINE1 siRNA#3	F: GGCCAUGGAACAAGGAUGATT	
R: UCAUCCUUGUUCCAUGGCCTT	
siRNA NC	F: UUCUCCGAACGUGUCACGUTT	
R: ACGUGACACGUUCGGAGAATT	

CCK-8 assay

Huh7 cell proliferation was quantified using the CCK-8 Cell Proliferation Detection Kit (KeyGEN BioTECH). Briefly, Huh7 cells transfected with either si-NC or si-SERPINE1 were cultured in 96-well plates for 48 h. Subsequently, 10 µL of CCK-8 reagent was added to each well, followed by incubation at 37 °C for 2 h. Absorbance values were measured at 450 nm using an ELx800 Microplate Reader (BioTek, Winooski, VT, USA).

Wound healing assay

Huh7 cells were seeded into six-well cell culture plates at a density of 1 × 105 cells/mL and incubated overnight. A sterile pipette tip was employed to create a scratch wound in each well. Unattached cells were removed by washing with 1× PBS, followed by the replacement of the culture medium with fresh medium. The cells were then concurrently transfected. After a 48-hour incubation period, images were captured at 100× magnification using an IX51 microscope (Olympus, Tokyo, Japan). Wound width was measured at 0 h (a) and 48 h (b), and the wound healing ratio [(a − b)/a × 100%] was calculated to assess migratory capacity.

Transwell assay

To assess Huh7 cell invasion, a 24-well transwell chamber (Corning Incorporated, USA) coated with Matrigel (BD, Franklin Lakes, NJ, USA) was employed. The cells were seeded at a density of 1 × 105 cells/mL within the transwell chamber. The lower chamber was filled with 500 µL of DMEM supplemented with 10% FBS. Following a 48-hour incubation period, the cells on the upper surface of the membrane were removed using cotton swabs. The cells on the lower surface of the membrane were stained with 0.1% crystal violet (Sigma) for 30 min at 37 °C, washed twice with 1× PBS, imaged using an IX51 microscope (Olympus, Japan), and quantified.

Total RNA extraction and RT-qPCR

Total RNA was isolated using TRIzol Reagent (Invitrogen, USA) according to the manufacturer’s protocol. RNA integrity was assessed via agarose gel electrophoresis, and concentration and purity were determined using a Nano100 spectrophotometer (Hangzhou Allsheng Instruments Co., Ltd., Hangzhou, China). First-strand cDNA synthesis was performed using the PrimeScript™ RT reagent Kit (Takara, Shiga, Japan) with total RNA as a template. For miRNA expression analysis, reverse transcription was conducted with a Bulge-Loop™ miRNA RT-PCR Starter Kit (RiboBio). GAPDH or U6 served as the endogenous control gene. Quantitative PCR was performed using SYBR Green PCR Mix (Takara) on a StepOnePlus Real-Time PCR System (ABI, USA). Relative gene mRNA and miRNA expression levels were calculated using the 2−ΔΔCt method based on three biological replicates with three technical replicates each. The primers sequences for miR-642a-3p, miR-3135a, miR-449b-5p, miR-642a-3p, miR-3135a, SERPINE1, GAPDH, and U6 were synthesized by General Biosystems (Anhui) Co., Ltd. (Anhui, China) and are listed in Table 2.

Table 2 The primer sequences of the genes.

Name	Sequence (5′ → 3′)	
miR-449b-5p	F:ACACTCCAGCTGGGAGGCAGTGTATTGTTA	
R:TGGTGTCGTGGAGTCG	
miR-544b	F: ACACTCCAGCTGGGACCTGAGGTTGTGCAT	
R: TGGTGTCGTGGAGTCG	
miR-642a-3p	F: ACACTCCAGCTGGGAGACACATTTGGAGAG	
R: TGGTGTCGTGGAGTCG	
miR-2116-3p	F: ACACTCCAGCTGGGCCTCCCATGCCAAGA	
R: TGGTGTCGTGGAGTCG	
miR-3135a	F: ACACTCCAGCTGGGTGCCTAGGCTGAGACT	
R: TGGTGTCGTGGAGTCG	
SERPINE1	F: GGTGCTGGTGAATGCCCTCTAC	
R: TGCTGCCGTCTGATTTGTGGAA	
GAPDH	F: AGATCATCAGCAATGCCTCCT	
R: TGAGTCCTTCCACGATACCAA	
U6	F: CTCGCTTCGGCAGCACA	
R: AACGCTTCACGAATTTGCGT	

Western blotting

Total protein extraction and quantification were performed using a total protein extraction kit (KeyGEN BioTECH) and a BCA protein content detection kit (KeyGEN BioTECH), respectively, according to the manufacturer’s protocols. As described by previous study (Chehade et al., 2021), the protein samples were separated by sodium dodecyl sulfate-polyacrylamide gel electrophoresis (SDS-PAGE) and subsequently transferred to polyvinylidene fluoride (PVDF) membranes. Immunoblotting was conducted using primary antibodies against SERPINE1 (ab222754, Abcam), E-cadherin (ab76055, Abcam), N-cadherin (66219-1-Ig, Proteintech), vimentin (bsm-33170m, Bioss), and GAPDH (ab9485, Abcam) at dilutions of 1:1000, 1:1000, 1:2000, 1:1000, and 1:2000, respectively. A secondary anti-rabbit IgG H&L antibody (ab6721, Abcam) was used at a dilution of 1:5000. Protein bands were visualized using an enhanced chemiluminescence (ECL) detection kit (KeyGEN BioTECH) and a ChemiDoc Touch 1708370 imaging system (Bio-Rad, USA). Densitometric analysis was performed using ImageJ software.

Dual-luciferase reporter assay

The 293T cells were seeded into 12-well plates. Upon reaching approximately 50% confluency, pmirGLO-SERPINE1-WT or pmirGLO-SERPINE1-MUT recombinant plasmid was co-transfected with miR-642a-3p mimics or negative control into 293T cells using Lipofectamine™ 3000 transfection reagent (Invitrogen, USA). After a 48-hour transfection period, 50 µL of cell lysate was transferred to each well of a 96-well black plate. Subsequently, Dual-Glo® Luciferase Reagent (Promega, USA) was added, and firefly luciferase luminescence was quantified using a Tecan Spark microplate reader (Tecan, Männedorf, Switzerland). Finally, 100 µL of Dual-Glo® Stop & Glo® Reagent (Promega, Madison, WI, USA) was added to each well to measure Renilla luciferase luminescence. Relative luciferase activity was normalized to Renilla luciferase activity.

Animal model

Four- to six-week-old male BALB/c nude mice obtained from Shanghai Lingchang Biotechnology Co., Ltd. were housed in single cages. The study protocol was approved by the Institutional Animal Care and Use Committee of Nanjing Ramda Pharmaceutical Co., Ltd. (IACUC-20230505) and carried out in accordance with the guidelines of the Animal Care Committee. Mice were acclimatized for one week under standard specific pathogen-free (SPF) conditions with a temperature of 20–26 °C, relative humidity of 40–70%, and a 12-hour light/12-hour dark cycle. Ten mice were randomly assigned to two groups: NC and shmiR-642a-3p (n = 5). Under abdominal anesthesia, nude mice were positioned supine, and the surgical site was disinfected. A midline abdominal incision was made to expose the liver, and the liver lobe exterior to the incision was gently removed with a cotton swab. Tumor cells were injected into the liver parenchyma approximately three mm deep using a needle, delivering 100 µL of Huh7-luc cells (Shanghai Zhong Qiao Xin Zhou Biotechnology Co., Ltd., China) at a density of 2 × 108 cells/mL. The liver was repositioned, and the abdomen was closed layer by layer. One week post-inoculation, mice in the NC and shmiR-642a-3p groups received intraperitoneal injections of 2 × 1011 vg of AAV-vector or AAV-shmiR-642a-3p, respectively. The animals (n = 5 per group) were monitored twice weekly for behavioral changes, food consumption, and weight. At eight weeks, one mouse from each group was anesthetized with carbon dioxide and subjected to liver tumor nodule examination (Fig. S1). The remaining mice were euthanized with carbon dioxide, and their livers were excised and photographed. Liver tissues were divided: one half was fixed in 4% paraformaldehyde, while the other half was snap-frozen in liquid nitrogen and stored at −80 °C.

Hematoxylin-eosin (H & E) staining

Liver tissues were fixed in a 4% paraformaldehyde solution and subsequently embedded in paraffin. Tissue sections with a thickness of 4 µm were prepared and stained with H&E. Histomorphological analysis of each liver was conducted using a SLIDEVIEW VS200 research slide scanner (Olympus).

Immunohistochemistry (IHC) assay

Livers were immediately immersed in 4% paraformaldehyde solution and fixed overnight. Tissue blocks were embedded in paraffin and sectioned to a thickness of 4 µm. Immunohistochemical staining for Ki67 expression in liver tissue was performed using the EnVision two-step method. Rabbit anti-Ki67 (ab16667; Abcam) at a dilution of 1:50 served as the primary antibody. Subsequently, sections were incubated with a specified HRP-conjugated secondary antibody (MXB Biotechnologies, Fuzhou, China). Diaminobenzidine (DAB) solution (MXB Biotechnologies) and hematoxylin (Nanjing Jiancheng Bioengineering Institute, Nanjing, China) were used for color development. Ki67 expression in liver tissue was visualized using a SLIDEVIEW VS200 research slide scanner (Olympus).

Statistical analysis

Data were analyzed using SPSS version 21.0 and presented as mean ±  standard error (SE) with at least three times. Independent-sample t-tests were employed to compare differences between two groups, while one-way analysis of variance (ANOVA) was used to assess differences among multiple groups. Post hoc comparisons were conducted using the Tukey method. All experiments were repeated at least three times. A P-value less than 0.05 was considered statistically significant.

Results

CAF-induced SERPINE1 underexpression promoted proliferation and invasion of Huh7 cells

CAFs are pivotal contributors to tumor progression, exhibiting a diverse range of functions encompassing collagen deposition and immunosuppression. Our prior investigation revealed that CAFs stimulated the proliferation and migration of Huh7 cells. To elucidate whether CAFs regulate Huh7 cells via SERPINE1, we initially quantified SERPINE1 mRNA expression in Huh7 cells co-cultured with CAFs. The results indicated a significant decrease in SERPINE1 mRNA expression within the co-culture group relative to the control group (Fig. 1A). Subsequently, SERPINE1 gene knockdown markedly enhanced Huh7 cell proliferation and invasion (P < 0.001) (Figs. 1B–1D), suggesting that CAFs may promote Huh7 cell proliferation and invasion through the suppression of SERPINE1 expression.

Figure 1 CAF-induced SERPINE1 knockdown promoted Huh7 cell proliferation and invasion.

(A) SERPINE1 mRNA expression in Huh7 cells was measured by RT-qPCR. (B) RT-qPCR was used to detect the expression of SERPINE1 mRNA following 48 h of SERPINE1 siRNA transfection. (C) The proliferation of Huh7 cells was detected by CCK-8 assay. (D) The invasion of Huh7 cells was detected by the transwell chamber. Magnification: 20×. Data are presented as the mean ± standard error (n = 3). *P < 0.05, **P < 0.01, and ***P < 0.001.

CAFs inhibited SERPINE1 expression in Huh7 cells by secreting miR-642a-3p

MicroRNAs function as intercellular messengers, transmitting information between cells, tissues, and organs. Within the TME, miRNAs contribute to tumor initiation and progression by regulating aberrant gene expression. To investigate whether CAFs influence SERPINE1 expression in Huh7 cells through miRNA secretion, we identified 18 potential miRNAs targeting the SERPINE1 3′ UTR through multiple miRNA online databases (Fig. 2A). Subsequent screening revealed five miRNAs of interest, among which miR-642a-3p, miR-3135a, and miR-449b-5p exhibited significantly elevated expression in the co-culture group (P < 0.05) (Figs. 2B–2F). Moreover, miR-642a-3p and miR-3135a expression was markedly increased in Huh7 cells with SERPINE1 knockdown (P < 0.01), with miR-642a-3p demonstrating a more pronounced difference between groups (Fig. 2G). These findings collectively suggest that CAFs suppress SERPINE1 expression in Huh7 cells by secreting miR-642a-3p.

Figure 2 MiR-642a-3p might bind to SERPINE1.

(A) Putative miRNAs binding to SERPINE1 3′ UTR using multiple miRNA databases, including StarBase, miRwalk, TargetScan, and miRDB. (B–F) Huh7 cells and CAFs were co-cultured in the transwell system for 48 hours, and the miRNA expression levels were measured by RT-qPCR. (G) MiR-642a-3p and miR-3135a expression in Huh7 cells following 48 h of SERPINE1 siRNA transfection. Data are presented as the mean ± standard error (n = 3). *P < 0.05, **P < 0.01, and ***P < 0.001.

CAFs-derived miR-642a-3p promoted migration, invasion, and EMT of Huh7 cells by inhibiting SERPINE1

To investigate whether CAF-derived miR-642a-3p promotes Huh7 cell migration, invasion, and EMT by regulating SERPINE1, miR-642a-3p mimics or inhibitors were transfected into Huh7 cells. Results demonstrated that miR-642a-3p mimics significantly upregulated miR-642a-3p expression while downregulating SERPINE1 mRNA expression (P < 0.001), whereas the miR-642a-3p inhibitor exhibited the opposite effect (Figs. 3A and 3B). Moreover, miR-642a-3p mimics significantly enhanced Huh7 cell migration and invasion, while the miR-642a-3p inhibitor significantly suppressed these processes (Figs. 3C and 3D). Notably, miR-642a-3p mimics markedly increased N-cadherin and vimentin protein expression and decreased E-cadherin protein expression in Huh7 cells (P < 0.001) (Fig. 3E). Conversely, the miR-642a-3p inhibitor significantly upregulated E-cadherin protein expression and downregulated N-cadherin and vimentin protein expression (P < 0.001) (Fig. 3E), suggesting that CAF-derived miR-642a-3p promotes EMT in Huh7 cells.

Figure 3 MiR-642a-3p promoted migration, invasion, and EMT of Huh7 cells.

(A–B) After 48 h of miR-642a-3p mimics or inhibitor transfection, miR-642a-3p and SERPINE1 mRNA expression in Huh7 cells were detected using RT-qPCR. (C) Cell migration ratio was measured by wound healing assay. Magnification: 10×. (D) Huh7 cell invasion was detected by transwell assay. Magnification: 20×. (E) Protein expression was examined by western blot. Data are presented as the mean ± standard error (n = 3). ns: P > 0.05, *P < 0.05, **P < 0.01, and ***P < 0.001.

To investigate the binding interaction between miR-642a-3p and the 3′ UTR of SERPINE1, recombinant plasmids pmirGLO-SERPINE1-WT and -MUT were constructed based on predicted binding sites from a miRNA database. A dual-luciferase reporter assay demonstrated a significant decrease in fluorescence activity in the WT group upon miR-642a-3p mimic overexpression (P < 0.05), whereas no effect was observed in the MUT group (P > 0.05) (Fig. 4A). Rescue experiments revealed that SERPINE1 knockdown attenuated the inhibitory effects of miR-642a-3p knockdown on cell migration, invasion, and EMT in Huh7 cells (Figs. 4B–4D), suggesting that miR-642a-3p promotes migration, invasion, and EMT in Huh7 cells by targeting SERPINE1.

Figure 4 MiR-642a-3p knockdown inhibited migration, invasion, and EMT of Huh7 cells by targeting SERPINE1.

(A) The binding of miR-642a-3p with SERPINE1 3′ UTR was detected by dual-Luciferase reporter assay. (B) After miR-642a-3p inhibitor or miR-642a-3p inhibitor combined with SERPINE1 siRNA was transfected into Huh7 cells for 48 h, the cell migration ratio was measured by wound healing assay. Magnification: 10×. (C) Huh7 cell invasion was detected by transwell assay. Magnification: 20×. (D) Protein expression was examined by western blot. Data are presented as the mean ± standard error (n = 3). ns: P > 0.05, *P < 0.05, **P < 0.01, and ***P < 0.001.

miR-642a-3p knockdown inhibited tumor growth and EMT in vivo

To assess the in vivo impact of miR-642a-3p on tumor infiltration, an orthotopic tumor model was established by injecting Huh7 cells into the livers of nude mice. As depicted in Fig. 5A, tumors were observed in the NC group but absent in the shmiR-642a-3p group. While no significant weight difference was observed between groups, a slight increase in weight was noted for the shmiR-642a-3p group (Fig. 5B). Histopathological examination of the NC group revealed visible tumor lesions invading the hepatic parenchyma with a larger invasion area compared to the reduced invasion area observed in the shmiR-642a-3p group (Figs. 5C and 5D). Moreover, miR-642a-3p knockdown significantly suppressed miR-642a-3p expression while enhancing SERPINE1 expression (Fig. 5E). Notably, miR-642a-3p knockdown upregulated SERPINE1 and E-cadherin protein levels while downregulating N-cadherin and vimentin protein levels (Fig. 5F), indicating an inhibitory effect on tumor infiltration and dissemination.

Figure 5 MiR-642a-3p knockdown suppressed growth of orthotopic liver tumors.

(A) Nude mice and liver tissues in the NC and shmiR-642a-3p groups. Four animals were shown in each group (n = 5). (B) Growth curve of the nude mice. (C) The structures of the liver tissues were observed by HE staining (n = 3). Magnification: 2× and 10×. (D) Ki67 expressions were analyzed using immunohistochemistry (n = 3). Magnification: 10×. (E) Expression levels of miR-642a-3p and SERPINE1 gene were detected by RT-qPCR (n = 3). (F) Protein expression was analyzed using western blot (n = 5). *P < 0.05, **P < 0.01, and ***P < 0.001.

Discussion

The TME is a critical determinant of cancer initiation and progression (Ye et al., 2024; Yu, Huang & Guo, 2024). In the TME, a large number of CAFs are recruited and activated, thus affecting cancer progression (Jing et al., 2024; Wang et al., 2024). It has been reported that CAFs are activated, proliferate, and accumulate in over 80% of HCC cases (Affo, Yu & Schwabe, 2017). These activated CAFs exert carcinogenic effects through multiple mechanisms, including the secretion of soluble factors and exosomes, as well as extracellular matrix (ECM) remodeling (Yin et al., 2019; Ying, Chan & Lee, 2023). Our findings revealed a significant decrease in SERPINE1 mRNA expression in Huh7 cells co-cultured with CAFs. Subsequent investigations demonstrated that SERPINE1 knockdown markedly enhanced Huh7 cell proliferation and invasion. This suggests that CAFs may accelerate the development of HCC by causing SERPINE1 gene to express poorly in HCC cells.

The role of SERPINE1 in cancer remains controversial (Jin et al., 2020; Nam, Seong & Hahn, 2021; Zhu et al., 2020). In gastric cancer, SERPINE1 knockdown significantly inhibited cell proliferation, migration, invasion, and xenograft tumor growth (Chen et al., 2022). SERPINE1 has been directly linked to EMT, tumor cell stemness, and chemoresistance in head and neck squamous cell carcinoma, with its overexpression correlating with increased metastasis risk (Pavón et al., 2016). Interestingly, SERPINE1 has been shown to promote senescence in lung cancer cells (A549 and H1299), thereby inhibiting tumor progression (Wang et al., 2023a). However, conflicting findings indicate that SERPINE1 upregulation promotes lung cancer cell invasion (Kong et al., 2021). Within the context of HCC, SERPINE1 is predominantly considered an oncogene, although some studies suggest an anti-cancer role. Jin et al., (2020) reported significantly higher SERPINE1 expression in HCC tissues compared to adjacent non-cancerous tissues, with a negative correlation between SERPINE1 expression and overall survival, suggesting its prognostic value (Jin et al., 2020). SERPINE1 has been shown to promote proliferation, migration and invasion in HepG2 cells (Li et al., 2021b). Conversely, our findings demonstrate that SERPINE1 knockdown enhances proliferation and invasion in Huh7 cells, indicating the heterogeneous nature of SERPINE1 expression and function within HCC cells. Notably, CAF-derived SERPINE1 exhibits tumor-suppressor activity in Huh7 cells.

MiRNAs are single-stranded, non-coding RNAs. They regulate gene expression by binding to the 3′ UTR of target messenger RNAs (mRNAs), leading to mRNA degradation or translational inhibition (Ali Syeda et al., 2020; Diener, Keller & Meese, 2024). Each miRNA can regulate multiple target genes, influencing a diverse array of biological processes, including differentiation, development, proliferation, migration, and apoptosis (Ali Syeda et al., 2020; Nemeth et al., 2024). In recent years, miRNAs have emerged as promising biomarkers for tumor diagnosis (Calis, Mogulkoc & Baltaci, 2020; He et al., 2020; Li et al., 2021a). Dysregulation of miRNAs, such as miR-155 (Vo et al., 2012), miR-541 (Xu et al., 2020), miR-126 (Zailaie & Sergi, 2022), and miR-17-5p (Zhou et al., 2024), has been implicated in the malignant progression and poor prognosis of HCC. These miRNAs influence critical HCC processes, including proliferation, apoptosis, metastasis, and drug resistance (Mallela et al., 2024). Moreover, miRNAs function as signaling molecules facilitating intercellular communication, enabling information exchange and gene regulation between tumor cells and other cell types, including CAFs and immune cells, ultimately contributing to tumor progression (Barrera et al., 2023; Qi et al., 2022; Salah et al., 2022). For instance, CAF-derived exosomal miR-20a-5p promotes HCC progression through the LIMA1-mediated β-catenin pathway (Qi et al., 2022), while CAF-derived exosomal miR-1228-3p enhances the resistance of liver cancer cells to sorafenib (Zhang, Pan & Shao, 2023).

Our study revealed that miR-642a-3p expression increased concurrently with a decrease in SERPINE1 expression in Huh7 cells co-cultured with CAFs, suggesting a potential inhibitory effect of CAFs on SERPINE1 expression in Huh7 cells via secreting miR-642a-3p. While limited research has explored miR-642a-3p in cancer, existing studies indicate its role in promoting tumor invasion, metastasis (Cao et al., 2022), and drug resistance (Qin et al., 2017; Yu et al., 2019). To investigate the involvement of CAF-derived miR-642a-3p in HCC progression through targeting SERPINE1, we confirmed the binding of miR-642a-3p to the SERPINE1 3′ UTR using a dual-luciferase reporter assay. Subsequent in vitro studies demonstrated that CAF-derived miR-642a-3p promotes HCC cell migration, invasion, and EMT by targeting SERPINE1. Moreover, in vivo experiments revealed that miR-642a-3p knockdown significantly suppressed tumor proliferation and dissemination in the liver, highlighting its critical role in HCC progression. Given the established role of exosomes in miRNA-mediated intercellular communication (Sheng et al., 2024; Zhang et al., 2024), future investigations will focus on determining whether CAF-derived miR-642a-3p is encapsulated in CAF-secreted exosomes and elucidating the functional implications of exosomal miR-642a-3p in CAF-HCC crosstalk.

At present, our research still has some limitations. While our study confirms that SERPINE1 knockdown promotes proliferation of the HCC cells, we did not explore the effect of CAF-derived miR-642a-3p/SERPINE1 axis on the cell proliferation, necrosis, or apoptosis, which is the most direct demonstration of the effect on tumors. Next, we will evaluate the role of miR-642a-3p/SERPINE1 axis in HCC cell proliferation, necrosis, and apoptosis using CCK-8, LDH cytotoxicity, and Annexin V/PI apoptosis assays. In addition, we will examine the expression of miR-642a-3p and SERPINE1 to assess their clinical implications for HCC staging and prognosis, using HCC tissue microarray technology. More significantly, using transcriptome, proteome, and metabolome sequencings, we will investigate the molecular mechanism of miR-642a-3p/SERPINE1 axis in multiple HCC cell lines to provide new insights for diagnosis and treatment of HCC.

Figure 6 Molecular pattern of CAFs-drived miR-642a-3p supporting the migration, invasion, and EMT of hepatocellular carcinoma by targeting SERPINE1.

Created with Figdraw.

In conclusion, CAF-derived miR-642a-3p promotes hepatocellular carcinoma cell migration, invasion, and EMT by targeting SERPINE1 (Fig. 6), suggesting its potential as a molecular marker for HCC treatment. Additionally, our study enriches the intricate functions of SERPINE1 in HCC.

Supplemental Information

Supplemental Information 1 The ARRIVE guidelines 2.0: author checklist

Supplemental Information 2 MIQE checklist

Supplemental Information 3 RT-qPCR analysis of SERPINE1 mRNA levels in Huh7 cells after coculture with CAFs

Supplemental Information 4 RT-qPCR analysis of SERPINE1 mRNA levels in Huh7 cells after SERPINE1 siRNA

Supplemental Information 5 RT-qPCR analysis of miR-642a-3p, miR-3135a, miR-449b-5p, miR-2116-3p, and miR-544b levels in Huh7 cells after coculture with CAFs

Supplemental Information 6 RT-qPCR analysis of miR-3135a and miR-642a-3p levels in Huh7 cells after SERPINE1 siRNA

Supplemental Information 7 RT-qPCR analysis of miR-642a-3p and SERPINE1 mRNA levels in Huh7 cells after miR-642a-3p mimics or inhibitor

Supplemental Information 8 The full-length uncropped gels/blots of SERPINE1 and EMT-related protein expressions in Huh7 cells after miR-642a-3p mimics or inhibitor

Supplemental Information 9 The full-length uncropped gels/blots of SERPINE1 and EMT-related protein expressions in Huh7 cells after miR-642a-3p inhibitor or miR-642a-3p inhibitor combined with si-SERPINE1

Supplemental Information 10 Nude mice and liver tissues in the NC and shmiR-642a-3p groups

Supplemental Information 11 RT-qPCR analysis of miR-642a-3p and SERPINE1 mRNA levels after miR-642a-3p knockdown

Supplemental Information 12 The full-length uncropped gels/blots of SERPINE1 and EMT-related protein expressions after miR-642a-3p knockdown

Supplemental Information 13 Effect of miR-642a-3p knockdown on growth of orthotopic liver tumors

One animal were shown first in each group (n = 5).

The authors would like to express their gratitude to Nanjing Ramda Pharmaceutical Co., Ltd. for providing the animal experiment platform and technical support.

Abbreviations

CAFs cancer-associated fibroblasts

HCC hepatocellular carcinoma

SERPINE1 serpin family E member 1

ncRNAs non-coding RNAs

miRNAs microRNAs

CCK-8 Cell Counting Kit-8

RT-qPCR real-time quantitative polymerase chain reaction

EMT epithelial-mesenchymal transition

NASH non-alcoholic steatohepatitis

HBV hepatitis B virus

HCV hepatitis C virus

TME tumor microenvironment

TAMs tumor-associated macrophages

PVDF polyvinylidene fluoride

ANOVA one-way analysis of variance

SE standard error

ECM extracellular matrix

UTR untranslated region

Additional Information and Declarations

Competing Interests

Author Contributions

Animal Ethics

Data Availability

The authors declare there are no competing interests.

Shuo Zhang conceived and designed the experiments, performed the experiments, analyzed the data, prepared figures and/or tables, authored or reviewed drafts of the article, and approved the final draft.

Gang Cao performed the experiments, prepared figures and/or tables, authored or reviewed drafts of the article, and approved the final draft.

Shuijie Shen analyzed the data, authored or reviewed drafts of the article, and approved the final draft.

Yu Wu analyzed the data, prepared figures and/or tables, and approved the final draft.

Xiying Tan performed the experiments, prepared figures and/or tables, and approved the final draft.

Xiaoyan Jiang conceived and designed the experiments, authored or reviewed drafts of the article, and approved the final draft.

The following information was supplied relating to ethical approvals (i.e., approving body and any reference numbers):

The study protocol was approved by the Institutional Animal Care and Use Committee of Nanjing Ramda Pharmaceutical Co., Ltd. (IACUC-20230505) and carried out in accordance with the guidelines of the Animal Care Committee.

The following information was supplied regarding data availability:

The raw data are available in the Supplementary Files.

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
