# Peer review of "CAF-derived miR-642a-3p supports migration, invasion, and EMT of hepatocellular carcinoma cells by targeting SERPINE1"

_PeerJ, doi:10.7717/peerj.18428_

## Round 0.1 · original submission · Major Revisions

the paper needs to be modified according to the reviewer's suggestions in order to improve the paper and make it suitable for publication. In section matherial and methods; for the western blot method, it is necessary to add a reference at the end of the paragraph

Reviewer 1 ·

Basic reporting

1. Some reference information is incomplete (line 386, 433, 446).
2. The manuscript contains many grammatical and syntax errors and must be carefully edited.

Experimental design

1. How many mice did the author use for the study? The number of mice described in the experimental method (line 190, n= 5 per group) is inconsistent with Figure 5 (line 529, n= 4 per group).
2. The transfection conditions (amount of plasmid, siRNA concentration, time point of harvesting cells… etc.) should be clearly described in the Methods.
3. The legend of figures is too brief and needs more detailed description.

Validity of the findings

In this study, the authors investugated the role of miR-642a-3p and SERPINE1 in the crosstalk between CAFs and HCC. They found that miR-642a-3p level is significantly increased in Huh7 cells cocultured with CAFs, which downregulates the expression of SERPINE1, thereby promoting the proliferation and invasion of Huh7 cells. The authors suggests that CAFs may inhibit the expression of SERPINE1 in Huh7 cells through secreting miR-642a-3p, and it can be used as a molecular marker for treatment of HCC.
In general, the interpretation of the data is reasonable, and the findings are interesting and may contribute to the treatment.
However, as described in the article, the role of SERPINE1 in cancer has been controversial. The opposite findings regarding its role in HCC have also been reported. The author only used one cell line to conduct the study, which may not be convincing enough to prove the role of SERPINE1 in HCC. The authors should perform experiments with more cell lines to provide stronger evidence.

Reviewer 2 ·

Basic reporting

This research article submitted to PeerJ, titled “CAFs-drived miR-642a-3p supports migration, 1 invasion, and EMT of hepatocellular carcinoma cells
by targeting SERPINE1” by Zhang et al., 2024

1. BASIC REPORTING
• Clear, unambiguous, English language,
• Title perfectly reflects the idea, disease, marker and the axes.
• The Keywords Section; ok,
• Abstract appropriately summarize the manuscript, but too concise, needs more discussion and info,
• No discrepancies between the abstract and the manuscript remainder,

Intro & background to show context but need to be clarified more and more info

Figures are relevant, high quality, well labelled & described.


• The introduction is showing less details, but needs more organization and clarification, and details and citations as several sentences with no ref.
line 46 the authors should add other causes of HCC as liver viruses as hepatitis C virus

• line 52-54, add a ref. to the sentence “Therefore, exploring biomarkers of HCC occurrence and development is highly important for prevention, diagnosis, treatment, and prognosis.”

• line 55 to 62 couldn’t rely only on one ref.
• Every sentence should have its ref.
• The references are old the newest is 2022 and for 2023 only 2 ref. and no 2024
• Need more about info and intro about ncRNA then miRs and then miRs in cancer generally as in breast cancer

Experimental design

• The purpose of the study is clearly defined, but, more clearer aims and objectives statement before the methods.
• The objectives should not state the results as the sentence implies “In vitro and in vivo studies revealed the role of miR-642a-3p/SERPINE1 axis in HCC cell migration, invasion and epithelial-mesenchymal transformation (EMT), providing a new target for the treatment of HCC.”

• Methods part: ok
• Statistical analysis ok but indicate the number of replicates
• The author defined terms used in the remainder of the manuscript,
• Tables and Figures are appropriate, appropriately labeled, with underneath legend to provide a clearer explanation, as well as the supplementary figures.
• Ethics for animal must identify ARRIVE
• The bioinformatics platforms need to be identified in a separate point not in the results to appear for the first time

Validity of the findings

• Discussion ok better than the introduction, however, several of the recommended ref. to add to be used in the discussion part as well.
• The discussion needs more elaboration.
• line 346 in the discussion stated a fact that needs a ref per this fact is not the author’s one “miRNA involvement in intercellular communication mainly depends on exosomes.”
• The sentence for such facts needs to be in the uncertain voice may or might be not “mainly depends on”
• “limitation(s)” are to be added not only the clinical samples but lacking the crisper experiments for proof of the concept
• The study lacks the prior in silico analysis
• The study should be followed by functional enrichment analysis for the downstream proteins, genes and the molecular docking for the serpine protein
• Conclusion(s) needs to be rewritten
• Mention also the “strength(s)” of the study,
• Mention “recommendation” or “future prospective” for completion of the work?
• List of abbreviations are needed.
• The final recommendation Reconsider after Major Revision.

---

## Round 0.2 · Minor Revisions

The work is potentially interesting

However, some more changes are necessary in order to fix the work

1. In the introductory part about the tumor-stoma interaction, the authors put only their own references, as well as many references exclusively by Chinese authors, as if no one else had done research on that topic. that is why it is necessary to add paper(s) that show the role of mediators and cells in tumor stroma interaction in order to confirm the importance of the described phenomenon

2. For the western blot determination method, at the end of the paragraph, add papers that also showed this earlier in order to confirm the significance of the method

3. In the legends of the images attached to the work, it is necessary to write the magnification of the microscope

4. the limitation of the study should be written especially for the findings obtained by the cell damage method that was used, and the findings obtained by observation with an ordinary microscope, which is now called the Wound healing assay, and in essence does not explain or show anything.

5. Today there are many other more specific methods such as determining the enzyme LDH and its passage through the cell membrane is much more significant and precise for describing cell necrosis

Reviewer 1 ·

Basic reporting

No comment.

Experimental design

No comment.

Validity of the findings

No comment.

Additional comments

The authors have addressed all questions raised during the first reviewing procedure. I found the manuscript has been significantly improved and recommended to accept.

Reviewer 2 ·

Basic reporting

Still the manuscript is not novel and couldn't be promoted for publication

Experimental design

Still the manuscript is not novel and couldn't be promoted for publication

Validity of the findings

Still the manuscript is not novel and couldn't be promoted for publication

---

## Round 0.3 · accepted · Accept

In my view, all the issues pointed by the reviewers and all constructive critiques were adequately addressed, and the manuscript was amended accordingly. Although the reviewer recommended rejection, I respectfully disagree with this recommendation, since novelty is not among the acceptance criteria in this journal. Therefore, I am accepting the revised manuscript in its present form.

Reviewer 2 ·

Basic reporting

I have stated previously to reject

Experimental design

I have stated previously to reject

Validity of the findings

I have stated previously to reject

Additional comments

I have stated previously to reject